# One Size Does Not Fit All: A Comprehensive Clinical Approach to Reducing Suicidal Ideation, Attempts, and Deaths

**DOI:** 10.3390/ijerph16193606

**Published:** 2019-09-26

**Authors:** David A. Jobes, Samantha A. Chalker

**Affiliations:** Department of Psychology, The Catholic University of America, Washington, DC 20064, USA; 97chalker@cua.edu

**Keywords:** suicidal risk, suicide-focused clinical care, stepped care

## Abstract

While the existence of mental illness has been documented for centuries, the understanding and treatment of such illnesses has evolved considerably over time. Ritual exorcisms and locking mentally ill patients in asylums have been fundamentally replaced by the use of psychotropic medications and evidence-based psychological practices. Yet the historic roots of mental health management and care has left a certain legacy. With regard to suicidal risk, the authors argue that suicidal patients are by definition seen as mentally ill and out of control, which demands hospitalization and the treatment of the mental disorder (often using a medication-only approach). Notably, however, the evidence for inpatient care and a medication-only approach for suicidal risk is either limited or totally lacking. Thus, a “one-size-fits-all” approach to treating suicidal risk needs to be re-considered in lieu of the evolving evidence base. To this end, the authors highlight a series of evidence-based considerations for suicide-focused clinical care, culminating in a stepped care public health model for optimal clinical care of suicidal risk that is cost-effective, least-restrictive, and evidence-based.

## 1. Introduction to the Problem

Suicide is a major public health issue around the world that accounts for almost 800,000 deaths per year [1]. In the United States suicide is the 10th leading cause of death with approximately 47,000 total deaths in 2017 and 1.4 million American adults attempted suicide in that same year [2]. While suicidologists and public health officials are understandably preoccupied with suicide deaths and suicide attempts, Jobes and Joiner [3] have recently reflected on the massive population of people who experience suicidal ideation and all too often escape the attention of our suicide prevention research, clinical treatments, and even national health care policies. In the United States, 10,600,000 American adults experience serious suicidal thoughts [4]—a worrisome cohort which dwarfs the populations of those who attempt and die by suicide.

To fully address the many challenges to clinical suicide risk reduction we will consider: the history of mental health care and its legacy for suicidal patients, the notion of mindsets about how to best help care for suicidal people, various contemporary developments that may be changing mindsets about clinical suicide prevention, the historic pursuit of suicidal typologies, evidence-based suicide-focused treatments, and finally a stepped care public health model.

## 2. History of Mental Health and Suicidal Patients

The history of the field of mental health and the treatment of suicidal patients is rather sordid and includes many disturbing developments over the years. Prior to European enlightenment, the mentally ill were largely understood to be deviants possessed by the devil and (or) evil spirits; religious exorcisms were frontline “treatments” [5]. It is interesting to note that some form of ritual exorcism exists across the major world religions (e.g., Christianity, Judaism, Hinduism, and Islam). It was not until the latter 18th century that mentally ill “lunatics” began to be understood and treated with more compassion as patients with an illness. The innovative French doctor Phillippe Pinel famously ordered in 1795 that mentally ill patients be released from their chains at a large asylum named la Salpetriere outside of Paris. This launched the notion of “moral treatment,” which helped change perceptions about mental illness and how such people with these illnesses should be treated [6]. Unfortunately, the continued association of mental illness with “asylums” of this era and in the years that followed does not seem either enlightened or particularly moral.

Early “treatments” of the mentally ill were crude and physically harmful (e.g., bloodletting and trephination—the drilling of holes in the cranium) [5]. Various methods of inducing seizures or comas were explored and collectively referred to as shock therapies in the early 20th century [7]. There was extensive experimentation using electricity that ultimately resulted in the development of electro-convulsive therapy (ECT), which in the present day has been found to be helpful as a last resort for patients with treatment-resistant, severe depression [8]. Massive doses of insulin were repeatedly administered to patients with schizophrenia to induced comas—insulin coma therapy (ICT) [9]. Behavior-altering surgeries such as lobotomies and cingulotomies were used often in the mid-20th century to control patient behavior [5].

### 2.1. Psychopharmacology for Suicidal Risk

Psychiatric care of the mentally ill took a notable turn in the 1950s with the advent of first generation anti-psychotic medications. The evolution of psychotropic medications has been extensive and has come to shape the prevailing assumptions about suicidal patients—that treating a mental disorder is the key to reducing the symptoms of suicidal ideation and behavior.

While medications have undoubtedly helped many who suffer with mental disorders, there is extensive evidence that targeting and treating mental disorders has little or mixed impact on suicidal risk [10,11,12]. Despite the widespread use of medication there is fairly limited data (based on randomized controlled trials—RCTs) about the efficacy of medicine on suicidality [13]. For example, the emergent use of ketamine on suicidal ideation for a few days [14] has short-lived effects for some suicidal patients. There is meta-analytic support for lithium carbonate among suicidal bipolar patients [15] and one un-replicated RCT has shown that clozapine can reduce suicidal ideation and attempts among thought-disordered patients [16]. Data on the effectiveness of anti-depressants with suicidal risk are quite mixed [10]. Notably, there are now three meta-analyses showing that treating mental disorders has little to no impact on suicidal ideation and behavior [10,11,12]. Notwithstanding the lack of evidence, prominent experts (e.g., [17]) insist on the primacy of treating mental disorders to reduce suicidal risk, even trivializing the effectiveness of suicide-focused psychological treatments that have been proven to work by replicated data from rigorous randomized controlled trials.

### 2.2. Legacy of Mental Health

So, what is the legacy of our history of managing and treating mentally ill people? On the one hand, humanity has been able to move from superstition and fantastic explanations for abnormal behavior to a more clinical and interventive approach. On the other hand, the legacy has created a “doctor knows best” mentality marked by a custodial, coercive, and paternal model that largely centers on controlling behaviors, by force if necessary. Relevant to our present consideration of the suicidal patient, a further remnant of this history is the idea that suicidal people belong in the hospital and that being suicidal is by definition “crazy,” so we must therefore treat this particular form of insanity. There are of course consequences to such considerations. Indeed, there is evidence that negative views of inpatient psychiatric care and fear of being hospitalized may compel suicidal patients to not be forthright with their clinical provider about their suicidal thoughts [18]. Moreover, insisting on the pre-eminence of treating mental disorders (particularly using only medicine) for suicidal risk defies the extensive evidence-base that supports the effectiveness of targeting and treating suicidal ideation and behaviors independently of psychiatric diagnoses (refer to [19]).

## 3. A Fixed Mindset about Suicidal Patient Care?

To be balanced and fair, we agree that countless suicidal people have likely been helped by inpatient psychiatric care and psychotropic medications. Nevertheless, the previously discussed lack of suicide-specific evidence is rather striking. To this end, Jobes has argued that some contemporary providers may make assumptions about the presumed effectiveness of inpatient care and the use of medicine on suicidal risk [20]. Importantly, such presumptions can have a major impact on the patient’s clinical disposition, and even the course of their entire life. What is particularly concerning is that some clinicians may find themselves not always working in the patient’s best interest due to countertransference issues or fear of litigation should a patient take their life. In turn, such issues and fears may lead to overly defensive practices (e.g., hospitalizing a patient who has passive suicidal thoughts). Moreover, such practice behaviors may be shaped by wishful thinking that a three to six-day hospital stay or that treating the mental disorder with medicine is actually more effective for suicidal risk than alternative approaches that are supported by the data.

Thus, there may be a misguided notion that a “one-size” approach (i.e., a brief hospitalization and medication to treat the disorder) will work for all suicidal patients [21]. Another way of understanding the possible insistence by some that these approaches are effective can be explained by Stanford psychologist Carol Dweck’s [22] notion of a psychological “mindset.” Dweck’s empirical work has shown the existence of two distinct mindsets: a “fixed” mindset versus a “growth” mindset. Her research shows how these mindsets are reliably associated with different outcomes for personal and professional success, and a growth mindset is much more adaptive and linked to successful outcomes.

Given this line of thinking, is it possible that many mental health providers have developed a certain fixed mindset about what is best for a suicidal person? It is hard for anyone to say with certainty, but in our view, the effective assessment and treatment of suicidal risk requires a growth mindset so that we are better able to embrace suicide-focused approaches that are supported by RCT evidence. Also, we must collectively develop a growth mindset at the public health policy level to fully appreciate that a “one-size” approach to suicidality does not sufficiently address the worldwide challenge of suicidal risk [21].

## 4. Key Developments That May Be Changing Our Mindset

There are various contemporary developments that may help workers in the field to move from a fixed mindset about hospitalization and medications for all suicidal individuals to a growth mindset, which is supported by the extant RCT evidence-base and enhanced and evidence-based clinical practices, which can be further supported by progressive mental health policy.

### 4.1. Stabilization Planning

Perhaps one of the most important developments over the past 20 years in clinical suicidology has been the development and use of different versions of suicide-focused interventions that focus on stabilization planning for prospective acute suicidal crises. In marked contrast to the coercive and unfortunate use of “No-Harm Contracts” or “No-Suicide Contracts,” various stabilization planning interventions for suicidal outpatients are intuitively more compelling and have proven effective in clinical trial research. The best known of these interventions is the Safety Plan Intervention (SPI) developed by Drs. Stanley and Brown [23]. Widely adopted in the American Veterans Affairs and the U.S. Department of Defense healthcare systems, the SPI has also been adopted in the public and private sectors as an alternative to coercive contracts that focus on what a patient promises not to do (i.e., kill themselves) versus planning for what they will do within a suicidal, dark moment of crisis. The Safety Plan guides the patient through the steps of identifying triggers, self-coping techniques, distraction by others, reaching out for supportive help, reaching out to professional help, and securing lethal means. Many have clinically embraced the Safety Plan Intervention early on as an intuitively better option to coercive no-suicide contracts (despite the absence of empirical support for so doing). Relatively recently however, the superiority of the SPI over no-harm contracting for reducing suicide attempt behaviors was clearly demonstrated [24] in a large cohort-comparison study of suicidal U.S. military veterans, and additional randomized controlled trial data are now being conducted.

A conceptual “cousin” of the SPI is the “Crisis Response Plan” (CRP), which was first developed by Rudd, Joiner, and Rajab [25] and further elaborated and rigorously studied by Bryan and colleagues [26,27,28,29,30]. The CRP has the patient note on an index card, in their own written words, various triggers, coping strategies, resources, and oftentimes their reasons for living. Bryan and colleagues [28] performed a convincing RCT comparing the CRP to no-harm contracts and showed a significant effect on both suicidal ideation and suicide attempts, reducing the latter by 76% at the six-month follow-up assessment.

Another variation on this theme is the CAMS Stabilization Plan (CSP), which is developed in the initial session of the Collaborative Assessment and Management of Suicidality (CAMS) [31,32]. Within this therapeutic framework, the CSP emphasizes securing lethal means, which is followed by a list of coping strategies, resources for outreach, ways for decreasing isolation, and potential barriers to attending CAMS-guided clinical care. The CSP has not been independently studied outside of its use within CAMS, but it is a crucial tool that is routinely used within this evidence-based suicide-focused clinical treatment.

### 4.2. Caring Contact Follow Up

A rather stunning research development occurred when psychiatrist Jerome Motto had the idea of sending a “caring letter” to post-discharged psychiatric patients who refused to seek further mental health treatment. In their now famous RCT, Motto and Bostrom [33] found that sending a simple letter expressing concern and care every four months to patients post-discharge over five years caused a reduction in suicides, when compared to patients who did not receive caring letters. This elegantly simple study has been a transformative discovery for the field.

Motto’s seminal work has led to various replications using different forms of “caring contacts” that have involved different versions of the original Motto idea of using letters. Indeed, this simple, inexpensive and scalable intervention has been investigated using postcards, letters, emails, and text messages [34]. While some data have been mixed, a larger review of published caring contact trials found it to be generally effective in reducing suicidal behaviors [35]. Nevertheless, these authors noted the need for more rigorous caring contact RCTs. To this end, a recent RCT [36] with suicidal military personnel using caring contacts via text messages was compared to treatment as usual. The investigators found that those receiving caring contacts via text message were less likely to have any suicidal ideation and fewer attempts from baseline to 12 months. However, the likelihood or severity of suicidal ideation and the number of suicide risk incidents (i.e., hospitalization or medical evacuation) were not significantly different between groups.

### 4.3. Lived Experience Perspective

It is hard to estimate the impact of people who have “lived experience” with suicidal thoughts, attempts, and encounters with conventional mental health care. Among the earliest pioneers in this area were American Terri Wise and Australian Keith Harris. Marsha Linehan is perhaps the most famous person to poignantly describe her extensive experiences when she was a highly suicidal teenager [37]. In any case, there can be no question that the lived experience movement has had a significant impact on suicide prevention policy making [38], emerging clinical practices, and research (e.g., [39]).

#### Lived Experience Peer-Support Movement

The World Health Organization’s World Mental Health Survey determined that across 21 nations, a majority of individuals thinking, planning, and attempting suicide do not receive clinical treatments [40]. The major barriers for suicidal individuals to seek mental health care include low perceived need, attitudes to treatment (e.g., the wish to handle it on one’s own), and practical concerns (e.g., financial concerns). Given these considerations, there is a recognition of the need for other possible ways for suicidal individuals to relieve their suffering beyond traditional primary or psychiatric care [40].

A survey on how suicidal individuals cope with their suicidal thoughts showed overwhelmingly that talking with someone who was not a mental health professional was the primary response. Only 12% of respondents included talking to someone in the mental health profession [41]. Given these data, Alexander and colleagues [41] have advocated for education and support for family and peers as another line of intervention for loved ones in crisis. A desire for increased peer support services as a way to improve care was also noted among consumers who experienced a psychiatric emergency [42]. This research highlighted the desire for peer support to improve emergency care in a variety of situations including during physical restraint, being referred to a post-discharge peer support group, and assistance in securing post-discharge services. Some mental health policy advocates are now promoting various peer-support services as part of a compelling alternative to our contemporary current clinical practices within emergency services (refer to: https://crisisnow.com/#core_elements).

Generally speaking, those with lived experience have personally had suicidal thoughts, feelings, and/or engaged in suicidal behavior(s). Importantly, people with lived experience are also willing to share their experiences with others as they advocate for better mental health care and encourage others with lived experience to participate in their efforts to reform care for suicidal risk [43,44]. The emergence of this perspective is underscored by multiple national and international organizations who have devoted web link resources to support people with lived experience (e.g., the Suicide Prevention Resource Center, Zero Suicide, American Foundation for Suicide Prevention, National Action Alliance for Suicide Prevention, National Alliance of Mental Illness, Centre for Suicide Prevention, American Association of Suicidology, and International Association for Suicide Prevention).

At the national policy level in the U.S., the Suicide Attempt Survivors Task Force of the National Action Alliance for Suicide Prevention has published “The Way Forward: Pathways to Hope, Recovery, and Wellness with Insights from Lived Experience” [45]. This landmark report focuses on suicide prevention practices that are evidenced-based while also incorporating personal testimonies of those with lived experience.

Lived experience advocates are now being promoted in many diverse areas to assist in the prevention of suicide. For instance, those with lived experience have created web pages (e.g., NowMattersNow.org; LiveThroughThis.org; CrisisNow.org) to document their experiences and also provide help to those seeking alternative treatments [46,47,48]. Help that is provided on these websites includes video testimonies and skills (e.g., dialectical behavior therapy skills). Moreover, lived experience participants have been included in randomized controlled trials to provide added support to more traditional “face-to-face” talk therapy. One study that examined men who presented to the emergency department for self-harm demonstrated that research studies can readily include individuals with lived experience [49]. A community can be formed around such a research project to provide long-lasting support within patient-centered research to offer an innovative way to reach more high-risk individuals [49].

### 4.4. Suicide-Specific Policy Developments

Over the past twenty years in the United States, there have been some notable suicide-specific policies that have significantly changed suicide-related clinical practices. By their very nature, these policies are designed to shift practitioners from a status quo approach to handling suicidal risk to utilizing alternative practices that are largely driven by empirical data.

#### 4.4.1. Joint Commission Sentinel Event Alerts

The Joint Commission (TJC) accredits well over 21,000 healthcare settings across the United States. Because suicide-related fatal outcomes have been among the leading “sentinel events” (i.e., failures in care resulting in adverse outcomes), TJC has issued various Sentinel Event Alerts to notify accredited institutions that certain practices must change and be observed in accreditation site visits or possible sanctions may ensue. To the surprise of some within the healthcare industry, TJC issued a Sentinel Event Alert entitled “Detecting and Treating Suicide Ideation in all Settings” [50]. While the particular alert has been re-framed as “aspirational” (versus a required expectation), their intent is plain: take suicide seriously, identify the risk, and treat it.

#### 4.4.2. Zero Suicide

Inspired by the work of the Clinical Care Task Force of the National Action Alliance, the “Zero Suicide” policy initiative has been game changing in terms of an A-Z approach to raising the clinical standard of care across systems of care by developing the following: leadership, training, assessment, identification (assessment), engagement, treatment, transition, and improvement [51,52]. While there has been some controversy connected to the name, there can be no arguing the abject success. Zero Suicide policies are embraced across the United States and now around the world. It is fair to say there is no policy initiative in the history of suicide prevention that has been more influential and impactful than Zero Suicide (see discussion by [38]).

#### 4.4.3. Recommended Standard Care

There has been little guidance about how to best meet clinical expectations for effective care of suicidal patients. In the United States, the Substance Abuse and Mental Health Services Administration (SAMHSA) sponsored a working group to develop affordable and evidence-based approaches to working with suicidal risk across outpatient, inpatient, and emergency department settings. The “Recommended Standard Care for People with Suicide Risk: Making Health Care Suicide Safe” document [53] recommends basic approaches to working with suicidal patients, primarily emphasizing: identification/assessment of risk, stabilization/safety planning, lethal means safety discussions, the National Suicide Prevention Lifeline, and caring contact follow-up (all addressed throughout this article).

## 5. The Pursuit of Suicidal Typologies

Since the birth of suicide research, the determined pursuit of suicidal typologies has been a major focus of the field. Perhaps the most notable initial attempt was by sociologist Emile Durkheim in his classic work *Le Suicide* in 1897 [54]. Durkheim posited that there were four distinct suicide typologies as a function of social integration: egoistic, altruistic, anomic, and fatalistic. One example of this model is a World War II soldier who heroically throws himself on a live grenade within combat to save the lives of his comrades in arms—a clear example of an altruistic suicide. Many psychological typologies have ensued over the following years. In recent times, acute and chronic states have been empirically established [55]. Advanced technology has been used in ecological momentary assessment (EMA) to identify six reliable and distinct patterns of suicidal thinking [56]. Latent profile analysis can be used to identify distinct types of suicidal patients [57]. Within the realm of diagnostic nosology, Joiner and colleagues have proposed a potential DSM-6 candidate diagnosis called “Acute Suicidal Affective Disturbance” [58]. Similarly, Galynker and colleagues [59] have proposed the “Suicide Crisis Syndrome.”

The pursuit of reliable typologies is particularly relevant when clinical treatments are considered. Indeed, Jobes argued many years ago for the pursuit of “prescriptive” treatments, that is, matching different interventions to different suicidal states [60]. The notion of routing certain suicidal patients to certain well-suited treatments was once considered a pipedream, however, the contemporary reality of this prospect is a central assertion within this article.

### Machine Learning

Another way to think about suicidal typologies is a rapidly emerging and exciting—albeit sometimes controversial—approach that is broadly referred to as “machine learning” (which is sometimes referred to as “big data” research). As described by Kessler, et al. [61], the goals of “precision medicine” are to understand how the effects of treatment are modified by patient characteristics and to develop “precision treatment rules” (PTRs) based on this understanding to determine which of the treatments under consideration is likely to yield the best outcome for each patient or fine-grained patient subgroup.

## 6. Effective Clinical Treatments for Suicidal Risk

As discussed elsewhere, there are over 80 randomized controlled trials (RCTs) of treatments where suicide-related outcomes are the primary focus [20,62]. Within clinical science, RCTs are the gold standard of what has proven effective in a causal manner because of their reliance on experimental designs and a-priori hypothesis testing. RCTs that have replicated results, particularly when results have been replicated by independent investigators rise to the top of the list when we consider empirically validated interventions. There are a number of interventions that have shown promise in a single RCT, for example, the Attempted Suicide Short Intervention Program (ASSIP) and Mentalization-Based Therapy [63,64]. However, we will focus on treatments that have been replicated and have independent RCT support. These include four distinct treatments/interventions that have been shown to effectively target suicidality.

### 6.1. Dialectical Behavior Therapy

The most notable and heavily researched treatment that has been shown to reduce suicidal behaviors regardless of the intent to die, is Dialectical Behavior Therapy (DBT) [65,66]. DBT has four main components: individual therapy, skills training, phone coaching, and a consultation team. DBT’s main goal is to teach the patient skills to regulate emotions and improve relationships with others (suicidality is always targeted at the forefront of care). Skills are taught through validation and acceptance with a genuine focus on behavioral change. DBT was one of the first evidence-based treatments shown to be effective in decreasing repetitive self-harm behaviors and suicide attempts. More recent results have demonstrated DBT’s continued impact on decreasing suicidal behaviors among high risk individuals such as those with borderline personality disorder [67], and decreasing suicide ideation [68] and self-harm [69] among adolescents. However, while DBT has shown impressive results in managing suicidal behaviors [70], it is not solely devoted to treating suicidality, and replicated results for reliably decreasing suicidal ideation are not consistent across all DBT RCTs.

### 6.2. Cognitive Therapy for Suicide Prevention

Another effective treatment that targets the “suicidal mode” is Cognitive Therapy for Suicide Prevention (CT-SP) [71]. CT-SP treats the clinical characteristics of suicidal behaviors [72] by using various cognitive therapy techniques, which have proven successful for treating an extensive array of psychiatric disorders [73]. In a well-powered RCT (with a deliberately longer follow-up period than previous RCTs—18 months), Brown and colleagues [71] found that patients in CT-SP treatment were 50% less likely to attempt suicide compared to those in the usual care treatment group. The researchers also found significant reductions in levels of depression and hopelessness in the CT-SP treatment group compared to the control. This study showed high internal validity; replication of the data in a real world setting (e.g., a community-based outpatient setting) with varied samples (e.g., those who have not attempted suicide, but with severe ideation) is a pending next step for the researchers of CT-SP [71].

### 6.3. Brief Cognitive Behavior Therapy

Brief Cognitive Behavior Therapy (BCBT) was used in one well-powered RCT with suicidal, active duty U. S. Army Soldiers and was shown to be effective for reducing suicide attempts [29]. As its name indicates, this modality is brief (i.e., 12 sessions) to accommodate short-term treatment environments. This variation of CBT suicide-focused care emphasizes: common effective treatment elements, developing skills (e.g., emotion regulation, mindfulness), a focus on the suicidal mode, and the development of self-management. Rudd and colleagues followed soldiers for 24 months [29] and found that compared to treatment as usual, those in the brief CBT group were 60% less likely to attempt suicide.

### 6.4. The Collaborative Assessment and Management of Suicidality

Jobes describes the Collaborative Assessment and Management of Suicidality (CAMS) as a distinctive therapeutic framework that targets suicidality [32]. As a framework, not a new psychotherapy, the CAMS intervention does not require clinicians to give up their theoretical orientation or abandon reliable techniques. Indeed, CAMS is a “non-denominational” approach where potentially any treatment can be used within the framework [31]. In RCTs comparing CAMS to usual care control conditions there is strong evidence that CAMS significantly reduces suicidal ideation [74,75,76] and overall psychiatric distress [74,76]; it also increases hope and retention to care [74]. In on RCT comparing DBT to CAMS, CAMS did as well as DBT in reducing suicidal attempts and self-harm behaviors [77]. Beyond the initial four published RCTs, five additional CAMS RCTs are in various stages of completion and will add to this growing body of literature.

## 7. A Stepped Care Public Health Model

Finally, because this journal broadly addresses public health issues, it is important to wrap up our discussion by focusing on some major considerations that impact clinical care before we discuss two different conceptual models for thinking about suicide-specific clinical care that might optimize treatment outcomes and thereby save more lives.

### 7.1. Suicidal People Who Do Not Seek Mental Health Care

In the investigation of the challenges posed by suicidal risk, one of the most striking issues is that the majority of suicidal people simply do not seek mental health treatment. Indeed, in their review of the literature Luoma et al. found that only 19% sought mental health care in the month prior to their death [78], while 32% sought mental health care in the year prior to their suicide. Using National Violent Death Reporting System data, Niederkrotenthaler and colleagues [79] found that only 38.5% suicide decedents were engaged in mental health care within two months of their death.

Furthermore, a sample of 198 suicidal people noted that “using the mental health system” was their #4 coping strategy (preceded by “spirituality/religious practices,” “talking to someone and companionship,” and “positive thinking”) [41]. Frankly, many suicidal people do not want to seek professional treatment because of their negative attitudes towards mental health care [80]. As Allen and colleagues have noted [42], when suicidal people do seek professional care (e.g., ED-based care), they want something different than what they get (e.g., a more humanistic and person-centered clinical response).

### 7.2. Matching Different Treatments to Different Suicidal People 

Despite the aforementioned concerns, we believe that the notion of prescriptive suicide-focused treatments is now increasingly possible. As shown in the lower half of Figure 1, the discussion begins with suicidal people who do not want to seek professional care. As we noted, this population make up the majority of the suicidal population. We can aspire to educate, provide access to the National Lifeline, and if they touch healthcare systems, perhaps we can endeavor to provide caring follow-up. However, beyond these largely public health-oriented approaches, we do not know exactly how to reach this group. Perhaps the evolving lived-experience perspective and peer-support movement can provide a more accessible and less stigmatizing approach for this population.

Considering the upper half of the Figure 1, for those suicidal people seeking mental health care, in an ideal world, machine-learning algorithms could be used to optimally route patients to proven treatments that are best suited to care for different suicidal states—a version of empirically-based prescriptive suicide-focused treatments. It should be noted that similar components (e.g., stabilization planning, lethal means safety, etc.) tend to be integral in each of these effective treatments. As a final note, while focusing on suicide attempts and deaths is important and necessary, we also need to bring more attention to the massive population of seriously suicidal ideators, who represent the figurative iceberg under the water of the suicide prevention challenge, if we want to reduce suicidal attempts and completions [3].

From an even larger perspective, Figure 2 (adapted from an earlier figure that appears in [81]) depicts a “stepped care” model for suicide care. The healthcare costs, represented on the y-axis, are probably the single biggest force shaping healthcare practices in the real world. To this end, the top of the pyramid notes the most expensive systems-level interventions (i.e., inpatient psychiatric hospitalization) down to the least expensive interventions nearer the bottom of the pyramid. Moreover, the bottom layer reflects the need to grow a massive paraprofessional community of caring people who are trained to work with people at risk. Jobes has therefore called for the development of a “National Mental Health Service Corps” (similar to the United States Peace Corps founded in the 1960s) that could create a large community of volunteers and (or) provide caring individuals who could serve in a range of capacities, such as screening and peer-based support with proper training and supervision [82]. There will never be enough clinical providers to meet the needs of 10.6 million adults with serious suicidal thoughts (in the United States example). Indeed, such personnel could provide much needed care on the National Lifeline, which is currently facing significant capacity issues.

As we move up the levels of systems within the pyramid, the use of different evidence-based treatments described in this article can be layered in each level of clinical care. Ultimately this stepped care model provides a way of thinking broadly to providing cost-effective, least-restrictive, evidence-based care for those at risk for suicide. If we truly aim to make a lifesaving difference, public and mental policy shaped by this kind of approach might make a meaningful difference by reducing suicides and suicide-related suffering in all its forms.

## 8. Summary and Conclusions

We have argued in this article that to move the field of mental health forward in terms of suicidal risk, we must move away from a “one-size-fits-all” approach to working with suicidal people. Rather, an approach that matches different evidence-based suicide-focused treatments (i.e., DBT, CT-SP, BCBT, and CAMS) to different suicidal states is clearly needed. We also need thoughtful conceptual models and progressive evidence-based policies (e.g., Zero Suicide) to optimally engage those suicidal people who do not seek care. Finally, an array of professional and paraprofessional approaches (e.g., including the support of those with lived experience) and various services are needed to better support those people who battle with suicidal thoughts, feelings, and behaviors.

## Figures and Tables

**Figure 1 ijerph-16-03606-f001:**
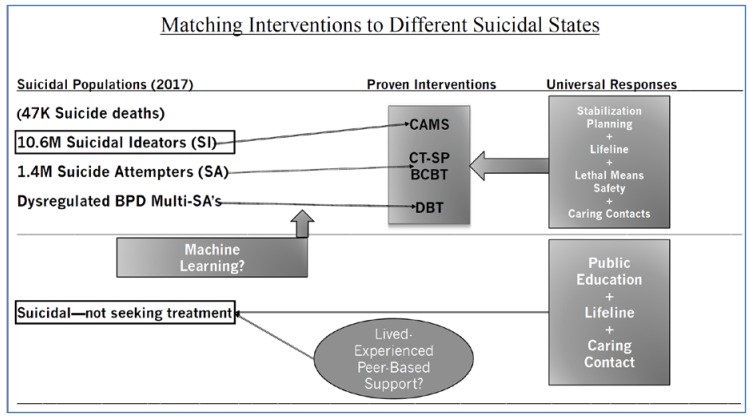
Different suicidal states include suicidal ideators (SI), suicide attempters (SAs), and multiple SAs. For those that seek mental health care: (SI) may be best matched to the Collaborative Assessment and Management of Suicidality (CAMS), SAs may be best matched to Cognitive Therapy for Suicide Prevention (CT-SP) or Brief Cognitive Behavior Therapy (BCBT), and dysregulated individuals with borderline personality disorder (BPD) with a multiple SA history may be best matched to Dialectical Behavior Therapy (DBT). Those that are suicidal but do not seek mental health treatment may be best matched to those with live-experienced and peer-based supports. Lifeline = the United States national suicide prevention lifeline phone number: 1-800-273-TALK (8255).

**Figure 2 ijerph-16-03606-f002:**
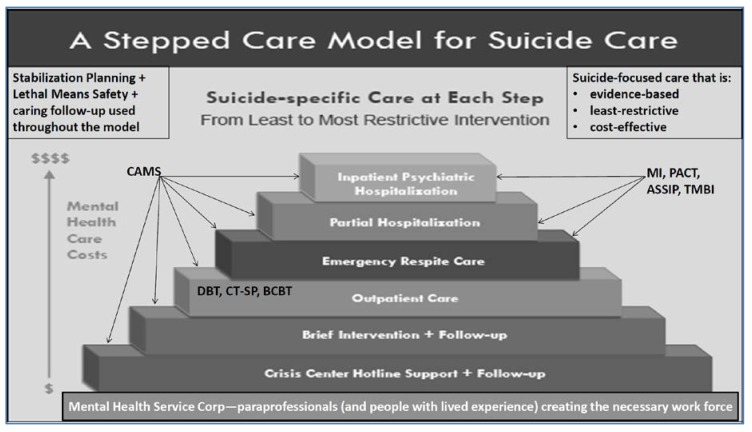
The y-axis is mental health care costs; the steps of the pyramid correspond from the bottom to the top with the least restrictive intervention the most restrictive intervention. ASSIP = Attempted Suicide Short Intervention Program; BCBT = Brief Cognitive Behavior Therapy; CAMS = Collaborative Assessment and Management of Suicidality; CT-SP = Cognitive Therapy for Suicide Prevention; DBT = Dialectical Behavior Therapy; MI = Motivational Interviewing; PACT = Post Admission Cognitive Therapy; TMBI = Teachable Moment Brief Intervention.

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
