# Peer review of "One Size Does Not Fit All: A Comprehensive Clinical Approach to Reducing Suicidal Ideation, Attempts, and Deaths"

_ijerph, 2019, doi:10.3390/ijerph16193606_

Round 1

Reviewer 1 Report

Title: One Size Does Not Fit All: A Comprehensive Clinical Approach to Reducing Suicidal Ideation, Attempts, and Deaths

The present manuscript represent a narrative review aimed at criticizing common clinical practice in the management of suicidal patients (such as hospitalization and medication) and describing several “suicide-specific” alternative approaches. Despite most of the findings exposed in the article are referred to US  initiatives and health policies, some of them can be adapted also to other countries.

The paper is interesting but I suggest that the Authors address the following issues:

A more structured objective paragraph is needed. Principal international guidelines on the management of suicide should be reported. Some medication are useful in the reduction of suicide risk (e.g. Suicide prevention strategies revisited: 10-year systematic review. Zalsman G et al. 2016). This should be better highlighted in the manuscript.

Author Response

Reviewer 1:

The present manuscript represent a narrative review aimed at criticizing common clinical practice in the management of suicidal patients (such as hospitalization and medication) and describing several “suicide-specific” alternative approaches. Despite most of the findings exposed in the article are referred to US  initiatives and health policies, some of them can be adapted also to other countries.

The paper is interesting but I suggest that the Authors address the following issues:

A more structured objective paragraph is needed.

We appreciate this reviewer's comment; however, the second paragraph (lines 35-39) of the introduction clearly lists our objectives for the manuscript. This paragraph has been highlighted in yellow within the manuscript to draw attention to it.

Principal international guidelines on the management of suicide should be reported.

We appreciate this comment; however, the intention of this manuscript was to focus on the United States and to not tackle the world-wide challenges which would be an entirely different paper altogether and not what we were asked to address when originally invited to contribute.

Some medication are useful in the reduction of suicide risk (e.g. Suicide prevention strategies revisited: 10-year systematic review. Zalsman G et al. 2016). This should be better highlighted in the manuscript. 

We agree with the reviewer’s comment. There is a paragraph (lines 66-78) in the manuscript that addresses the current data on medication for suicide risk. However, as explicitly stated in the manuscript, we have focused on randomized controlled trials (RCT’s) with replication trials by independent researchers (the highest level of scientific rigor). The article that the reviewer provided (Zalsman et al., 2016) supports the claim we have already made (lines 70-73) and we have added this systematic review as additional support for this argument.

Reviewer 2 Report

see the attached file for major revision

JournalIJERPH (ISSN 1660-4601)

Manuscript ID ijerph-594148

Type Review

Number of Pages 18

Title One Size Does Not Fit All: A Comprehensive Clinical Approach to Reducing Suicidal Ideation, Attempts, and Deaths

Authors David A. Jobes * , Samantha A. Chalker

The topic covered in the following article is of considerable interest both in clinical and research terms, some major revisions should be made.

The abstract must be structured according to the journal's guidelines, emphasizing the objectives and results and conclusions of the work.

In line 33 please specify the word “serious” suicidal thoughts. Do you mean high lethality, or severe intensity?

In general the session entitled Introduction to the Problem it is underdeveloped considering the importance of the topic while the second part is too long for an article of scientific interest.

History of Mental Health and Suicidal Patients: too long for a scientific paper.

The two figures should be made more clearly.

The second part of the paper is more scientific, however authors must enter newer bibliographic references.

If you read the article correctly, the purpose of the article is not so good, what are the goals that the authors set for themselves?

The conclusions are not well written, the results derived from the studies presented in the article are missing.

The limitations of the study are missing.

Author Response

The topic covered in the following article is of considerable interest both in clinical and research terms, some major revisions should be made.

The abstract must be structured according to the journal's guidelines, emphasizing the objectives and results and conclusions of the work.

We appreciate the reviewer’s suggestion; however, given that this is a review policy paper the structure of the abstract according to the journal's guidelines was followed to the our best ability since the paper is not typical scientific study reporting data from a new study.

In line 33 please specify the word “serious” suicidal thoughts. Do you mean high lethality, or severe intensity?

The word “serious” is the language SAMHSA uses in the report cited in the manuscript and cited below. Thus the report does not explicitly specify if “serious” reflects lethality or intensity.

[1] SAMHSA 2017 study (2018): Center for Behavioral Health Statistics and Quality, “Key substance use and mental health indicators in the United States: Results from the 2017 National Survey on Drug Use and Health (HHS Pub. No. SMA 17-5068, NSDUH Series H-53),” Substance Abuse and Mental Health Services Administration, Rockville, MD, 2018.

In general the session entitled Introduction to the Problem it is underdeveloped considering the importance of the topic while the second part is too long for an article of scientific interest.

We respectfully disagree with this reviewer’s comment. We think the introduction to the problem is well-developed given the intention and focus of the paper we were invited to write. This is intended to be a policy-driven paper not a pure science paper reporting new data that is solely focused on raising the standard of clinical care for suicidal patients across mental health.

History of Mental Health and Suicidal Patients: too long for a scientific paper.

We respectfully disagree with this reviewer’s comment as this is not intended to be a scientific paper reporting new data, but instead a policy driven paper. We believe this portion is necessary at the current length as it is indeed meant to specifically highlight the legacy left by early efforts to manage and treat suicidal risk with clear implications for contemporary issues we discuss for the balance of the ms. We have nevertheless added headers to this section to help make clear our intentions of this section while breaking up the information for easier reading.

The two figures should be made more clearly.

The two figures are as clear as we can make them be, particularly Figure 2 that is has been re-printed from a previous article. Incidentally we have obtained permission from the American Psychological Association to re-print and further modify this figure.

The second part of the paper is more scientific, however authors must enter newer bibliographic references.

With all due respect, we disagree and are puzzled by this comment. Over 72% of our references are from the last 5-10 years; almost 57% of our references are from the last 5 years. There are some historic references that pertain to historic material that we cover early on, but we feel that our work and citations are actually quite contemporary.

If you read the article correctly, the purpose of the article is not so good, what are the goals that the authors set for themselves?

Once again, we respectfully disagree. As noted previously, the goals or our paper are explicitly listed in the second paragraph (lines 35-39)--which is highlighted in yellow now for ease of reference.

The conclusions are not well written, the results derived from the studies presented in the article are missing.

Given the vast array and comprehensiveness of topics covered in this often conceptual/policy relevant paper--that also describes ground-breaking rigorous suicide-focused research--we respectfully disagree completely with this particular reviewer feedback.

The limitations of the study are missing.

Again, it would seem that this reviewer is critiquing a scientific paper reporting new data. The manuscript that we were invited to submit for this special section is the paper we have written. We therefore respectfully disagree with this point as we do not see a need for a limitation section that would otherwise be written for a scientific report of novel data. Our manuscript is not a study of new data but is more of a thought-piece, with policy implications, and related and thorough review of the related literature. In truth, a major goal of this paper is the highlight and describe a number of limitations for the field at large in terms of the existing standard of clinical care for suicidal patients, which in our view is often inadequate with life or death implications.

Reviewer 3 Report

This article focuses on a "one size does not fit all." This is a great concept.  I suggest researching psychopharmacology with the addition of psychotherapy in reducing suicides.  The best research is lithium or clozaril if we talk about medications and it's efficacy along with DBT for reducing suicide ideation.

Author Response

This article focuses on a "one size does not fit all." This is a great concept. 

I suggest researching psychopharmacology with the addition of psychotherapy in reducing suicides.  The best research is lithium or clozaril if we talk about medications and it's efficacy along with DBT for reducing suicide ideation.

We appreciate the reviewer’s positive appraisal. As for the comment about medication we once again agree that more research needs to be done with psychopharmacology in addition to psychotherapy to help clinically decrease suicidal risk.

Round 2

Reviewer 2 Report

The authors responded to every comment of the previous revision. The manuscript is improved.

Reviewer 3 Report

Great work with your revisions and contribution to this very important topic.  No more editing is required.